# EEG Dynamics of a Go/Nogo Task in Children with ADHD

**DOI:** 10.3390/brainsci7120167

**Published:** 2017-12-20

**Authors:** Simon Baijot, Carlos Cevallos, David Zarka, Axelle Leroy, Hichem Slama, Cecile Colin, Nicolas Deconinck, Bernard Dan, Guy Cheron

**Affiliations:** 1Department of Neurology, Hôpital Universitaire des Enfants Reine Fabiola, Université Libre de Bruxelles, 1020 Brussels, Belgium; sbaijot@ulb.ac.be (S.B.); nicolas.deconinck@huderf.be (N.D.); bernard.dan@ulb.ac.be (B.D.); 2Neuropsychology and Functional Neuroimaging Research Unit, Center for Research in Cognition and Neurosciences, Université Libre de Bruxelles, 1050 Brussels, Belgium; hichem.slama@ulb.ac.be; 3Cognitive Neurosciences Research Unit, Center for Research in Cognition and Neurosciences, Université Libre de Bruxelles, 1050 Brussels, Belgium; ccolin@ulb.ac.be; 4Laboratory of Neurophysiology and Movement Biomechanics, Université Libre de Bruxelles, CP640, 808 route de Lennik, 1070 Brussels, Belgium; ccevallo@ulb.ac.be (C.C.); dzarka@ulb.ac.be (D.Z.); Axelle.Leroy@ulb.ac.be (A.L.); 5Departamento de Ingeniería Mecánica, Facultad de Ingeniería Mecánica, Escuela Politécnica Nacional, Quito 170517, Ecuador; 6Research Unit in Osteopathy, Faculty of Motor Sciences, Université Libre de Bruxelles, 1070 Brussels, Belgium; 7Department of Clinical and Cognitive Neuropsychology, Erasme Hospital, Université Libre de Bruxelles, 1070 Brussels, Belgium; 8Laboratory of Cognitive and Sensory Neurophysiology, CHU Brugmann, Université Libre de Bruxelles, 1020 Brussels, Belgium; 9Medical and Rehabilitation Departments, Inkendaal Rehabilitation Hospital, 1602 Vlezenbeek, Belgium; 10Laboratory of Electrophysiology, Université de Mons, 7000 Mons, Belgium

**Keywords:** ADHD, Go/Nogo, EEG, brain oscillations, rhythms, ERP, ERSP, ITC

## Abstract

Background: Studies investigating event-related potential (ERP) evoked in a Cue-Go/NoGo paradigm have shown lower frontal N1, N2 and central P3 in children with attention-deficit/hyperactivity disorder (ADHD) compared to typically developing children (TDC). However, the electroencephalographic (EEG) dynamics underlying these ERPs remain largely unexplored in ADHD. Methods: We investigate the event-related spectral perturbation and inter-trial coherence linked to the ERP triggered by visual Cue-Go/NoGo stimuli, in 14 children (7 ADHD and 7 TDC) aged 8 to 12 years. Results: Compared to TDC, the EEG dynamics of children with ADHD showed a lower theta-alpha ITC concomitant to lower occipito-parietal P1-N2 and frontal N1-P2 potentials in response to Cue, Go and Nogo stimuli; an upper alpha power preceding lower central Go-P3; a lower theta-alpha power and ITC were coupled to a lower frontal Nogo-N3; a lower low-gamma power overall scalp at 300 ms after Go and Nogo stimuli. Conclusion: These findings suggest impaired ability in children with ADHD to conserve the brain oscillations phase associated with stimulus processing. This physiological trait might serve as a target for therapeutic intervention or be used as monitoring of their effects.

## 1. Introduction

Attention Deficit/Hyperactivity Disorder (ADHD) is a neurodevelopmental disorder affecting 3–7% of school-age children [1]. The main symptoms are age-inappropriate levels of inattention, impulsivity and hyperactivity. It is associated with school difficulties and social impairment [2]. Impairments of cognitive functions such as alertness, vigilance, effort, planning, organization and working memory have been documented [3,4,5]. Pathophysiology involves genetics [6,7] and neurobiological factors [8,9,10,11,12].

Neuroimaging studies have shown alterations in the neuronal connectivity [8,13,14] and brain electrical activity [15,16] in children with ADHD compared to typically developing children (TDC). Event-related potentials (ERP) have been used to approach the neural mechanisms underlying attention and inhibition deficits in children with ADHD. ERP components from visual tasks reflect both visual and condition-specific attentional processing. A variety of ERP features related to executive functions were found to be modified in children with ADHD compared to TDC, including N1 and P3 in selective attention, and N3 and P3 in response inhibition [17,18]. In particular, Go/Nogo or continuous performance task studies showed reduced frontal N1, frontal N2 and central P3 in children with ADHD [19,20,21,22]. More recently, developments of electroencephalographic (EEG) dynamics tools [23,24] have permitted to analyze specific event-related synchronization (ERS, increased power) or desynchronization (ERD, decreased power) in different frequency bands in the event-related spectral perturbation (ERSP) and inter-trial coherence (ITC) linked to sensory stimulation or behavioral events [25]. These frequency bands are classically decomposed as delta (1–4 Hz), theta (4–7 Hz), alpha (8–15 Hz), beta (15–30 Hz) and gamma (30 Hz and more). Several studies investigated EEG rhythm in children with ADHD compared to TDC during event-related task [26,27,28,29,30,31,32,33,34,35,36,37,38]. In particular, studies showed atypical delta activity in ADHD during continuous performance task (CPT) [27,30,37]. Yodanova et al. (2006) showed that late theta oscillation (200–450 ms) in response to irrelevant stimuli of auditory task was larger in children with ADHD than TDC [28]. In a study that investigated EEG familiarity in sibling pairs with ADHD, Loo et al. (2008) showed that theta and alpha oscillations were correlated with CPT response variability and omission errors [31]. Moreover, Nazari et al. (2011) showed that contrary to TDC, CPT induced alpha power increased in children with ADHD compared to eyes-open resting state [34]. Using a Sternberg memory task, Lenartowicz et al. (2014) showed a lower ERD during encoding phase and an increased alpha power during maintenance period in children with ADHD compared to TDC [35]. In previous studies about memory, Lenz et al. (2008, 2010) revealed an increased gamma-band response in children with ADHD than TDC during stimulus encoding of a memory task [32,33]. In the present study, we investigate EEG dynamics linked to the ERP triggered by sensory and behavioral events in a visual Go/Nogo paradigm in children with ADHD in order to dissociate oscillatory features of attentional orienting and motor/inhibition processing.

## 2. Materials and Methods

### 2.1. Participants

Twenty six children aged 8 to 12 years were enrolled in the study (14 with ADHD and 12 TDC). Children with ADHD were recruited consecutively from the pediatric neurology outpatient clinic of Hôpital Universitaire des Enfants Reine Fabiola (HUDERF) in Brussels, Belgium. Diagnosis of ADHD was made by multidisciplinary team including pediatric neurologists and neuropsychologists according to DSM IV-TR. If the child was treated by medication (methylphenidate), this was stopped 48 h before the experimental process. TDC were recruited from primary mainstream schools, and were screened by neuropsychologists of the team. In addition to the neuropsychological assessment, each child performed the Wechsler Intelligence Scale for Children (WISC-IV: Wechsler, 2005), and all parents were asked to fill in the Child Behavior Checklist (CBCL; [39]). Exclusion criteria were a seizure disorder, mental retardation (IQ below 80), psychiatric comorbidity, sensory deficits and pharmacologic treatment (other than methylphenidate) that could interfere with behavioral performances or with electrophysiological results. Five children were rejected due to mental retardation or psychiatric comorbidity. One child asked to stop the experiment because it was too long for him. Data from six children were rejected during data treatment due to too many artifacts. Thus, 14 children (7 in each group) were analyzed (ADHD: 2 boys and 5 girls, mean = 9.43 years; SD = 1.72/TDC: 4 boys and 3 girls, mean = 8.71 years, SD = 1.49). No difference between groups was found in age (*p* = 0.48, Mann-Whitney U Test), in gender ratio (Chi-square (1) = 1.17, *p* = 0.28), and comorbidities (Table 1). The experimental design was setup in agreement to CONSORT directives and the Helsinki declaration [40]. Informed consent was obtained from all participants and their parents. Experimental setup was approved by the Ethics Committee of CHU Brugmann and HUDERF (Brussels, Belgium).

### 2.2. Stimuli

The subject sat comfortably at 120 cm in front of a 17-in computer screen, and performed a visual-cued Go/Nogo task adapted from [41]. The Cue, Go and Nogo stimuli were displayed in black on a white background. Luminance and contrast of screen was the same during session and between sessions. Each cue (black square) was briefly shown (150 ms) and followed by a Go (“×”), or Nogo (“+”) stimulus during 150 ms after a 1–2 s random period of white screen. The task was divided in two blocks including 60 trials each, with equal probability of Go and Nogo stimuli. The participant had to press as quickly as possible a button after the Go stimulus, and retain/inhibit the pushing after the Nogo stimulus. The next Cue stimulus reappeared after 2.5 s after the Go/Nogo stimulus was displayed (Figure 1).

### 2.3. Behavioral Analysis

Regarding behavioral measures, dependent variables were omissions (no responses to Go trials), false alarms (“FA Cue”: pushing when the Cue is shown, and “FA Nogo”: pushing when “+” is shown), reaction time (RT) and the RT variability. The RT variability, estimating the intra-individual variability, was calculated with the coefficient of variation (CV) of reaction times [42], a normalized measure of dispersion, defined as the ratio of the standard deviation (σ) to the mean (μ): CV = σ/µ. The CV is useful because the standard deviation of data must be understood in the context of the mean of the data [43]. In order to highlight behavioral difference between groups we have cross-correlated to two time series corresponding to latencies of the Go and Button Press, in the ADHD and TDC groups respectively.

### 2.4. EEG Recording

Brain electrical activity was recorded with an ASA EEG/ERP system (ANT company, Enschede, The Netherlands) from 14 channels (Fz, F3, F4, Cz, C3, C4, Pz, P3, P4, Oz, O3, O4 and M1-M2 for the left and right mastoids), embedded in a waveguard cap (10–20 system), referred to the mean of the two mastoids and connected to the ground electrode. EEG activity was recorded at a sampling rate of 512 Hz (analog filtering: 0.1–100 Hz; amplification × 20). Eye movements were monitored using two bipolar recordings: one between each outer eye canthus, and one between a supra orbital electrode and an electrode just below the lower lid on the right side. Impedances were kept below 10 kΩ and checked before each recording. 

### 2.5. EEG Data Processing

Off-line data treatment and analysis were performed by means of EEGLAB software (SCCN, San Diego, USA) [23,44] and in-house MATLAB-based tools [45]. Band pass filter 0.1 to 125 Hz and notch filter around 50 Hz (47.5–52.5 Hz) and 100 Hz (97.5–102.5 Hz) were applied to attenuate electrical artifacts. Portions of data and defective electrodes (max. 6%) were removed by careful visual inspection. Ocular (blinks and saccades) and any other remaining artifacts (muscular, cardiac) were isolated and rejected by independent component analysis (ICA) on continuous data [46,47,48]. Data were cut in epochs extracted from −0.5 to 1 s of each stimuli onset. Only successful trials (button pressed after a GO and not pressed after a Nogo) were considered, rejecting all others. In all cases, epochs were rejected according to ±100 µV threshold criteria, and we made a visual review to confirm epoch rejection. After rejection, a total of 2217 epochs remained (ADHD: 47.9 ± 18.6 per subject and condition; TDC: 57.7 ± 20.2 per subject and condition).

### 2.6. EEG Data Analysis

A EEGLab design study was used for each stimulus to average data from all subjects. A time window of 500 ms before stimulus onset was used as baseline. Evoked potential (ERP), spectral perturbation (ERSP) and inter-trial coherence (ITC), time-locked on each stimulus (Cue, Go and Nogo), were calculated by averaging baseline corrected epochs extracted from −0.5 to 1 s of the stimuli onset. ERSP and ITC are both time–frequency measurements calculated by means of fast Fourrier transform (FFT), but the information they provide is different: ERSP measures variations in power spectrum at specific frequency ranges of ongoing rhythms at specific periods of time and frequency range [49,50]. A color code at each image pixel indicated the reached power (in dB) at a given frequency and latency relative to the stimulation onset. ERD (event-related desynchronization) indicates a power spectrum reduction, while ERS (event-related synchronization) indicates a power spectrum increase. ITC measures the synchronization across trials of ongoing oscillations (~phase locked) with respect to an event, independently of amplitude variation [51]. The ITC measure took values between 0 and 1. A value of 0 represented the absence of synchronization between EEG data and the time locking events; a value of 1 indicated their perfect synchronization. Statistical difference between ADHD and TDC groups were performed by EEGLab non-parametric permutation test at each trial latency of the average ERPs and every time-frequency point for ERSP and ITC [23]. Multiples comparisons are corrected by the False Discovery Rate (FDR) method [52]. Statistical differences of ERP, ERSP and ITC between groups were described by the electrodes of relevance, interval of time and amplitude (Table 2). For clarity, we subdivided the presentation of results by considering conditions and three regions: anterior (including Pz, P3, P4, Oz, O3, O4), central (Cz, C3, C4), and posterior (Fz, F3, F4).

## 3. Results

### 3.1. Omission, Commission & Reaction Time

There were more omissions to the Go stimulus in children with ADHD than TDC (*z* = −2.17, *p* = 0.029), but no difference was found in commission to Cue (*z* = 1.53, *p* = 0.125) and NoGo (*z* = 1.41, *p* = 0.160) stimuli. We noted no significant differences in time reaction between ADHD and TDC (RT_ADHD_ = 508.1 ± 83.3 ms, RT_TDC_ = 501.4 ± 50.5 ms; *z* = 0.383, *p* = 0.701). However, Figure 2 illustrates that the time of the button-press (BP) did not occur accordingly to the time of the Go in the ADHD group (Figure 2A) as in the case of TDC group (Figure 2B). No significant correlation between time of the Go and the time of the BP was obtained in the ADHD group, while a peak in the cross-correlation function at zero lag was obtained in the TDC group (91.8 ± 6.3%; Figure 2). In line with these results, children with ADHD showed more variation in their time reaction to a Go than TDC (CV_ADHD_ = 0.353 ± 0.087 s, CV_TDC_ = 0.245 ± 0.050 s; *z* = 2.43, *p* = 0.015).

### 3.2. ERP, ERSP & ITC Summarized in Table 2

#### 3.2.1. EEG Dynamics Responses to Cue Stimuli

Group analysis over the posterior regions (Figure 3) showed that Cue stimulus evoked a lower P1-N2 potentials (*p* < 0.01) in children with ADHD compared to TDC. In the frequency domain, we observed a smaller ITC of theta (4.5–7 Hz) oscillation, that reached significant threshold (*p* < 0.05) between 260 and 420 ms after the Cue stimulus. At the same latency, high beta-gamma frequency showed a lower power and higher ITC in ADHD compared to TDC. Over central regions, we showed no significant difference between group in ERP, as well as in frequency domain, in response to the Cue stimulus. Concerning frontal regions, group analysis showed a lower N1 and higher P2 amplitude in ADHD than TDC (*p* < 0.05). These were coupled by a lower alpha power (*p* < 0.05) and ITC (*p* < 0.05) in children with ADHD compared to TDC. 

#### 3.2.2. EEG Dynamics Responses to Go Stimuli

Concerning Go condition, ERP analysis showed a lower P1-N2 and lower P3 potential over posterior regions in children with ADHD compared to TDC (*p* < 0.05). In these regions, theta power and ITC were lower around 350 ms. Over central regions (Figure 4), we found a significantly lower P3 amplitude (*p* < 0.01) in children with ADHD compared to TDC. Moreover, while TDC showed an alpha (7–13 Hz) ERD starting at about 250 ms after the Go stimulus, this was not observed in children with ADHD (*p* < 0.05), and was coupled to a smaller ITC of alpha oscillation in ADHD than in TDC group (*p* < 0.01). Group analysis in frontal regions showed in children with ADHD compared to TDC, a lower N1 and higher P2 amplitude (*p* < 0.05) with at the same latency, a lower alpha power (*p* < 0.05). Finally, we revealed at about 600 ms in all regions, that gamma power was lower in children with ADHD compared to TDC. 

#### 3.2.3. EEG Dynamics Responses to Nogo Stimuli

In response to the Nogo stimulus, we found over posterior regions a lower P1-N2, with at the same latency a higher alpha power and lower alpha ITC in children with ADHD compared to TDC. Over central regions, although ADHD group seemed get a higher N3 than TDC, we observed no significant difference in ERP between groups. However, we found a lower theta power and ITC in children with ADHD compared to TDC at about 300 to 400 ms. Frontal N3 over anterior regions was lower in ADHD with respect to TDC group at about 300 ms (*p* < 0.01, Figure 5). At this latency, frequency domain showed a highly significant lower theta-alpha (5–11 Hz) power (*p* < 0.01) and ITC (*p* < 0.01) in ADHD group compared to TDC. Finally, we observed a significant high-beta/low-gamma ERD (*p* < 0.05) and a higher gamma ITC (*p* < 0.05) between 350 and 600 ms over all regions in children with ADHD, compared to TDC. 

## 4. Discussion

In this study we explored the functional links between behavioral measurements, ERP and EEG dynamic perturbations evoked by a Go/Nogo task in children with ADHD. Firstly, we observed the previously described reduction of the P1 and N2 potentials in posterior regions at about 100–200 ms [19,22,53,54]. However, in our results this occurred in all conditions (Cue, Go and Nogo), suggesting a relationship with early visual and/or attentional processes. Moreover, our data suggested for the first time that this ERP alteration recorded in the occipital region was related to a deficit in the theta ITC (i.e., the ability to phase-locked this ongoing oscillation). While the common finding is that theta activity was higher in ADHD compared to TDC at rest [35] and in an auditory selective attention task [28], our results point toward another type of deficit related to the ability to reorganize the phase of theta oscillations in a reproductive and coherent way along the different trials. Indeed, increased theta power at rest, generally coupled with decreased beta activity (theta/beta ratio, TBR), has been put forward as the most robust electrophysiological finding in ADHD [55]. However, it has been questioned by several studies reporting insufficient accuracy of theta power (46.8–63%) and TBR (40.3–58%) to discriminate children with or without ADHD [56]. In particular, the link between TBR and arousal in ADHD lacks consistency [55]. As an alternative view, our result suggests that increased theta power observed in many studies may have reflected a compensatory effect to a theta-phase disturbance. In term of state regulation deficit [57], this compensatory effect could be the substrate for effort allocation difficulties in children with ADHD. Theta oscillation is considered to be a basic physiological element involved in global oscillatory synchronization processes linking together multiple brain regions [58]. Such linkage would be at the origin of conscious perceptions [59] and offers an interesting model to address attentional processes and their perturbations.

Another finding in this study is the presence of high-beta/low-gamma ERD between 300 ms and 600 ms in ADHD group with respect to TDC. This was observed in response to all conditions, but more specifically to Go and Nogo conditions. The fact that the earliest and strongest gamma ERD and gamma ITC was recorded in the frontal region during the Nogo condition could be related to the frontostriatal circuitry considered as the initial target of ADHD study [60] and in particular to the dorsolateral and orbitofrontal cortices activated during the inhibitory control related to the Nogo [61,62]. This frontal gamma ERD recorded in ADHD children is consistent with the hypoactivation of the frontal cortex observed in ADHD during the Go/Nogo task [63], and may be accounted for by desynchronization and relative paucity of the firing of the cortical neurons involved in the gamma rhythm generation in children with ADHD compared to TDC. More generally, changes in gamma power were reported to be correlated with attentional modulation [64,65]. Recorded gamma ERD in response to the Go and Nogo stimuli suggests changes driven by lowered attention in alerted condition in children with ADHD over the trials course. This interpretation could relate to impaired working memory (WM) processes typically found in children with ADHD [32,33,35], and may indicate a basic deficit in the maintenance of the persistent activity of the WM mainly assumed by recurrent circuitry. Indeed, although the present task does not specifically assess WM as paradigms such as the ‘delayed match to sample with distractors task’ do (Miller et al., 1991), the Go/Nogo paradigm does require WM in order to keep the behavioral meaning of the three different visual items in memory. According to a recent computational framework of a WM task [66], the presence of beta-gamma oscillation implies the coexistence of a ground state and persistent activity in such a way that transient input (such as one of the three different items in this study) can initiate a persistent state (with a specific aim) corresponding to the gate-in mode of the dynamic gating regime. 

In this model, while the beta-gamma oscillation encodes the persistent activity of the WM, the theta band oscillations (3–8 Hz) encodes a selective-gating mode maintaining the information previously stored in memory [66]. Recent studies suggest that cross-frequency coupling such as theta-phase gamma-amplitude coupling (TGC) reflect cortico-subcortical interactions. TGC was first observed in rat hippocampus when gamma power occured accordingly to specific phase of theta rhythm during task [67]. It has been proposed that TGC plays a functional role in multi-scale neuronal communication between local domains of cortical processing and large-scale brain networks. It was involved in various cognitive processes such as visual perception [68], short-term memory [69], and learning [70]. In particular, Nakatani et al. (2014) showed in an attentional blink paradigm that the synchrony of fast oscillations (beta and gamma) with slow activity (theta and high delta) increased with improved efficiency in the task [71]. Kim et al. (2015) showed that children with ADHD exhibited lower TGC at rest in frontal, temporal and occipital areas than controls [72]. In another study of 68 children with ADHD in a continuous performance task (CPT), low performance was associated with high levels of synchronization between theta-phase and 40 Hz gamma power [73]. The authors suggested that abnormal augmentation of TGC reflects a dysfunctional interaction of the attention/arousal system at the multi-scale large network level [73]. In particular, theta phase and TGC perturbation could be explained by interference of default mode network during task which altered working memory. Silberstein et al. (2016, 2017) showed that children with ADHD exhibited increased functional connectivity over prefrontal and parieto-frontal regions that was not apparent in TDC, during the interval preceding the target appearance in a CPT [74]. The authors suggested that children with ADHD had difficulties suppressing inadequate cortical networks that may interfere with task completion [75,76]. Moreover, they suggested that default mode network (DMN) acting as top-down process was preferentially mediate by alpha and beta oscillations, while bottom-top process would be mediated by theta and gamma oscillations.

A last result from our data concerning Cue, Go and Nogo conditions concern a decrease of alpha power and ITC in frontal regions concomitant to a decrease of N1-P2 potentials in children with ADHD, compared to TDC. Alpha oscillation has been interpreted as reflecting global inhibition of the cortex, improving behavioral performance by facilitation of the cognitive control [77,78,79,80,81,82,83], and it has been suggested that it provides a gate-out mode facilitating WM task completion and memory cleaning [66]. Thus, alpha power may participate in enhancement of the signal-to-noise ratio in order to selectively update relevant incoming information [84], and access to memory [78]. From this perspective, decreased power of alpha oscillation over frontal regions in the ADHD group could be related to a disturbance of signal-to-noise cleaning effect, inducing some difficulties to avoid irrelevant information for task performance. In particular, Lenartowicz et al. (2014) showed decreased ERD of the alpha rhythm during memory encoding and increased ERS of the same rhythm during memory maintenance in children with ADHD with respect to TDC [35]. Otherwise, recent findings suggested that alpha-phase plays an important role for temporal attention. In particular, Gruber et al. (2014) showed that pre- and peri-stimulus alpha-phase alignment contributed to generated P1 and was associated with shorter target decision time [85]. In our results, time-frequency response to the Go in ADHD group was characterized by a lower theta-alpha phase-locking than TDC at about 200 to 300 ms, following by a higher alpha power from 200 to 600 ms. This was coupled with a lower P3 potential in children with ADHD compared to TDC, as classically described [18,41]. Moreover, accordingly to previous Go/Nogo studies [86,87], our behavioral results showed that reaction times were significantly more variable in children with ADHD than TDC. It has been proposed that these variations could be explained by a state regulation deficit in ADHD, suggesting an inability to adjust the internal state according to the needs of the situation [88]. From this perspective, variability in responding has been considered to reflect effort instability [57]. Children with ADHD were probably less able to allocate effort in a task to keep their performance at a relatively stable level during manipulation of event rate [88]. Taken together, these EEG and behavioral results suggest that variability of reaction time in ADHD would be related to a disturbance of theta and alpha phase-locking among the different trials. 

## 5. Conclusions

In conclusion, this study showed that ERP studies must be analyzed during EEG dynamics, in order to better approach the mechanisms that underlie neural processing in ADHD. In relation to this, we found, in children with ADHD compared to TDC, that in Cue, Go and Nogo conditions: (1) reduction of occipito-parietal P1-N2 components was coupled with a decrease of theta ITC (extended to a decrease of alpha ITC in Go and Nogo); (2) reduction of frontal N1-P2 components was coupled with a decrease of alpha power and ITC; (3) an overall decrease of high-beta/low-gamma power occured 300 ms after stimulus (this was coupled with an increase of gamma ITC in the Nogo condition); and (4) in the Go condition, the decrease of P3 was preceded by a high alpha power and a small alpha ITC. This points to impaired ability in conserving the phase of brain oscillations associated with stimulus processing in a Go/Nogo paradigm in various frequency bands. This physiological finding might serve as a target for therapeutic intervention, or be used as monitoring of their effects. This study also paves the way for further investigations comprising the high EEG density and source localization approach.

## Figures and Tables

**Figure 1 brainsci-07-00167-f001:**
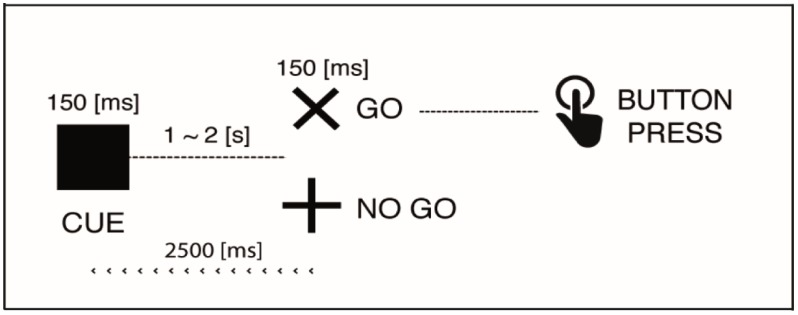
Go/Nogo paradigm.

**Figure 2 brainsci-07-00167-f002:**
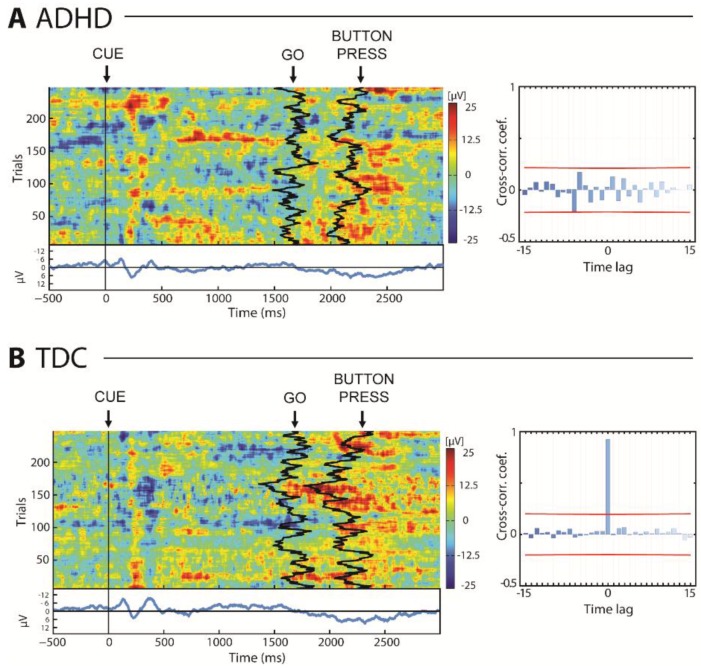
Reaction time in attention-deficit/hyperactivity disorder (ADHD) (**A**) and typically developing children (TDC) (**B**) groups, including cross-correlation between occurrence of Go and PB stimuli. Red line indicates significance *p* < 0.05.

**Figure 3 brainsci-07-00167-f003:**
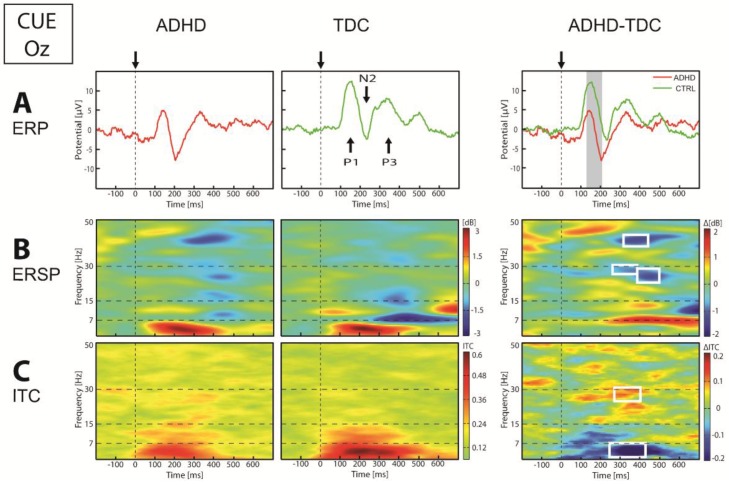
Event-related potentials (ERP) (**A**), event-related spectral perturbation (ERSP) (**B**) and inter-trial coherence (ITC) (**C**) in Oz electrode in response to Cue stimulus. On the ERP panel: arrows and vertical dotted lines indicate stimulus onset, vertical gray band indicates significance at *p* < 0.05. On the ERSP and ITC panel: vertical dotted lines indicate stimulus onset, horizontal dotted lines define frequency bands, and white square indicate significance at *p* < 0.05.

**Figure 4 brainsci-07-00167-f004:**
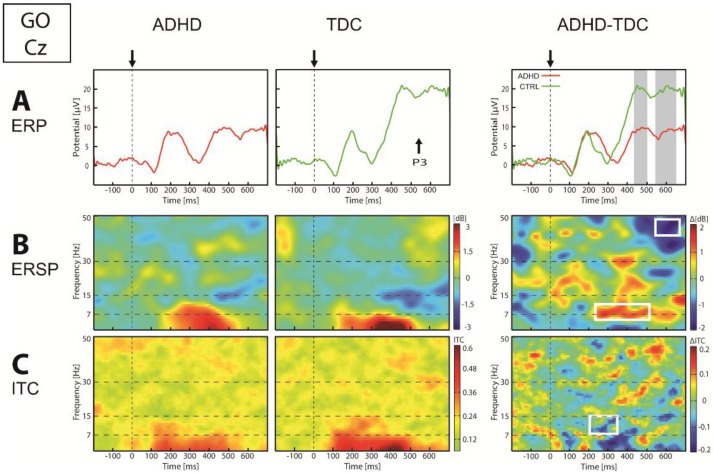
ERP (**A**), ERSP (**B**) and ITC (**C**) in Cz electrode in response to Go stimulus. On the ERP panel: arrows and vertical dot lines indicate stimulus onset, vertical gray bands indicate significance *p* < 0.05. On the ERSP and ITC panel: vertical dotted line indicate stimulus onset, horizontal dotted lines define frequency bands, and white square indicate significance *p* < 0.05.

**Figure 5 brainsci-07-00167-f005:**
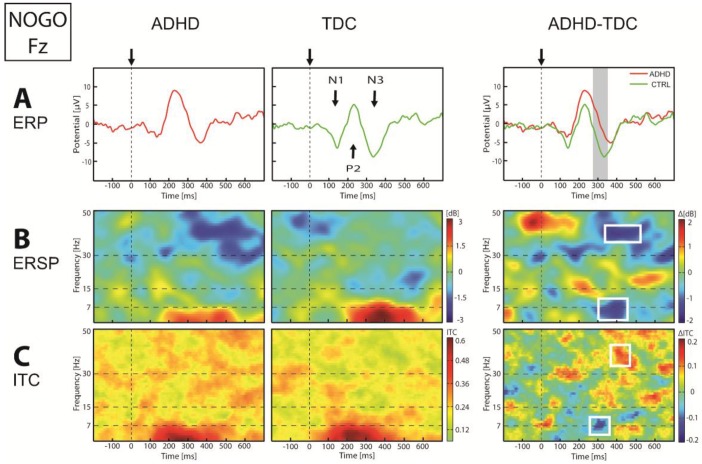
ERP (**A**), ERSP (**B**) and ITC (**C**) in Fz electrode in response to Nogo stimulus. On the ERP panel: arrows and vertical dotted lines indicate stimulus onset, vertical gray bands indicate significance *p* < 0.05. On the ERSP and ITC panel: vertical dotted line indicate stimulus onset, horizontal dotted lines define frequency bands, and white square indicate significance *p* < 0.05.

**Table 1 brainsci-07-00167-t001:** Group comparison for estimated IQ, age and parent-rated Child Behavior Checklist (CBCL) T-scores.

	W	*p*	*p* ^a^
Full-scale QI	21.50	0.747	
Affective Problems	30.00	0.007	0.04
Anxiety Problems	20.00	0.407	1
Somatic Problems	14.50	1.000	1
ADHD Problems	30.00	0.005	0.03
Oppositional Defiant Problems	28.50	0.017	0.11
Conduct Problems	28.50	0.09	0.11

Note, W: Wilcoxon Mann-Whitney U Test. ^a^
*p*-values below are corrected for multiple comparisons (Bonferroni correction).

**Table 2 brainsci-07-00167-t002:** Summary of results in children with ADHD compared to TDC.

ADHD–TDC						
Stim.	Regions	Measure	Comp./Freq.	Time (ms)	Electrodes	Sig. ^1^
Cue	Posterior Figure 3	ERP	↓P1-N2	115–205	Oz, O1, O2	*p* < 0.009
			↑N2	170–191	Pz, P3	*p* < 0.045
		ERSP	↓β (27–32)	238–485	Oz, O1, O2, P4	*p* < 0.044
			↓γ (40–43)	332–427	Oz, O2, O1	*p* < 0.049
		ITC	↓θ (4–8)	261–425	Oz, O1, O2, Pz, P3, P4	*p* < 0.039
			↑β (25–30)	273–422	Oz, O1, P4, P3	*p* < 0.044
	Anterior	ERP	↓N1	121–161	Fz, F3, F4, Fpz	*p* < 0.018
			↑P2	223–273	Fz, F3, F4, Fpz	*p* < 0.019
		ERSP	↓α (9–12)	150–312	Fz, F3, F4	*p* < 0.035
		ITC	↓α (10–13)	101–263	Fz, F4	*p* < 0.049
Go	Posterior	ERP	↓P1-N2	146–189	Oz, O1, O2, Pz, P3	*p* < 0.022
			↓P3	345–439	Oz, O1, O2, Pz, P3	*p* < 0.044
		ERSP	↓θ (4–7)	302–390	Oz, O1, O2, P3, P4	*p* < 0.034
			↓γ (42–47)	570–679	Oz, O1, O2, P3, P4	*p* < 0.032
		ITC	↓θ-α (5–12)	320–453	Oz, O1, O2, Pz, P3	*p* < 0.041
	Central Figure 4	ERP	↓P3	464–656	Cz, C3	*p* < 0.012
		ERSP	↑α (7–13)	252–541	Cz, C3, C4	*p* < 0.031
			↓γ (42–47)	566–593	Cz, C4	*p* < 0.032
		ITC	↓α (8–14)	220–347	Cz, C3, C4	*p* < 0.018
	Anterior	ERP	↓N1	117–128	Fz, F3, F4	*p* < 0.044
			↑P2	207–260	Fz, F3, F4	*p* < 0.011
		ERSP	↓α (10–18)	109–234	Fz, F3, F4	*p* < 0.003
			↓γ (38–47)	517–654	Fz, F4	*p* < 0.032
Nogo	Posterior	ERP	↓P1, ↑N2	136–209	Oz, O1, O2, Pz, P3	*p* < 0.005
		ERSP	↑α (12–16)	126–273	Oz, O1, O2, Pz, P3, P4	*p* < 0.003
			↓γ (31–36)	296–380	Oz, O1, O2, Pz, P3, P4	*p* < 0.018
		ITC	↓θ-α (5–14)	308–414	Oz, O1, O2, P3	*p* < 0.005
			↑γ (31–36)	302–410	Oz, O2, Pz, P3, P4	*p* < 0.007
	Central	ERSP	↓θ (4–8)	255–423	Cz, C3	*p* < 0.002
			↓γ (38–43)	428–509	Cz, C3, C4	*p* < 0.048
		ITC	↓θ (4–8)	291–369	Cz, C3	*p* < 0.016
			↑γ (31–37)	275–345	Cz, C3, C4	*p* < 0.041
	Anterior Figure 5	ERP	↓N1-P2	136–211	F3, F4	*p* < 0.049
			↓N3	279–344	Fz, F3, F4, Fpz	*p* < 0.018
		ERSP	↓θ-α (4–11)	277–396	Fz, F3, F4	*p* < 0.014
			↓γ (38–45)	316–517	Fz, F3, F4, Fpz	*p* < 0.022
		ITC	↓θ-α (5–11)	256–390	Fz, F3	*p* < 0.016
			↑γ (34–44)	382–511	Fz, F3, F4	*p* < 0.003

Note: Arrows indicated increase or decrease of ERP amplitude or ERSP/ITC in children with ADHD compared to TDC. ^1^
*p*-values below are corrected for multiples comparisons (False Discovery Rate method). ERP: Evoked-Related Potentials; ERSP: Evoked-Related Spectral Perturbation; ITC: Inter-trials Coherence.

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
