# Peer review of "EEG Dynamics of a Go/Nogo Task in Children with ADHD"

_brainsci, 2017, doi:10.3390/brainsci7120167_

Round 1

Reviewer 1 Report

The present study of EEG dynamics and coherence during a Go/NoGo task in 14 ADHD and 12 typically developing children (8-12 years) is of interest in terms of objective measurement of cognitive (physiological) deficits in ADHD children. While the number are relatively small, the demonstration of lower theta/alpha inter-trial coherence (ITC) in ADHD children is not new but the demonstration that this occurred across trials is new. The investigators relate this to a dynamic “gate-in” effect, which is impaired in ADHD children, and relate this impairment to a working memory deficit. The discussion posits an energy deficit but does not mention findings by Silberstein et al (Brain and Behaviour, 2016; Biological Psychiatry, 2017) in relation to default mode interference as an alternate model. Minor edits include: Line 157 should read ‘did not occur’ Line 310 should read ‘In relation to this...’ Line 316 should read ‘This points…’

Author Response

We thank the reviewer for his/her positif comments. Following the proposed suggestions, we have now included the work of Silberstein et al. (2016, 2017) about fonctional connectivity abnormalities in ADHD, and difficulties of these children to suppress inadequate default mode network that may interfere with task completion.

We have taken into account these constructive suggestions. We hope that after the modification that we have made, the paper can be now considered for publication.

Reviewer 2 Report

This manuscript reports Research on EEG dynamics of a Go/Nogo task in children with ADHD. The research is interesting however, authors need to address some of the concerns before the paper is considered for publication.

The paper is well written but in order to further improve the manuscript I suggest authors to do the following;

1) Please add some more information regarding the similar studies conducted in either introduction or literature section.

2) This study uses EEG for data analysis and interpretation but authors have not mentioned artifacts removal procedure related to the study. Since the study uses 14 EEG channels the artifact removal procedure is important. Techniques such as ICA/BSSS are widely used for the same. Please cite the following papers related to artifact removal.

S. Bhardwaj et al, "Online and automated reliable system design to remove blink and muscle artefact in EEG," 2015 37th Annual International Conference of the IEEE Engineering in Medicine and Biology Society (EMBC), Milan, 2015, pp. 6784-6787.

Feis, Rogier A., et al. "ICA-based artifact removal diminishes scan site differences in multi-center resting-state fMRI." Frontiers in neuroscience 9 (2015).

P. N. Jadhav et  al, "Automated detection and correction of eye blink and muscular
artefacts in EEG signal for analysis of Autism Spectrum Disorder,"
2014 36th Annual International Conference of the IEEE Engineering in Medicine and Biology Society, Chicago, IL, 2014, pp. 1881-1884.

3) Rationale behind choice of non-parametric permutation test as compared to other similar statistical measures need to be explained.

4) Please explain the anterior and posterior results in detail.

5) Finally, if there are any similar studies then please compare your results with other studies.

Author Response

We thank the reviewer for all his/her comments and suggestions.

1) We have added in the introduction section a review of ADHD studies investigating frequency bands during an event-related task (Yodanova et al. 2001, 2006, 2013, Johnstone et al. 2003, Alexander et al. 2008, Loo et al. 2008, Lenz et al. 2008, 2010, Nazari et al. 2011, Lenartowicz et al. 2014, Rommel etal. 2017).

2) We thank the reviewer for providing all these relevant references. Indeed we have used the ICA procedure (from EEGlab) for blink, saccade, muscular and cardiac artifact rejection. Following the proposed suggestions, we have added the next references: Bhardwaj et al. (2015), Feis, Rogier A., et al. (2015), and to Jadhav et  al. (2014).

3) Non-parametric permutation was used considering the small sample size. In practise,  permutation test is one of the two tests (the other one is bootstrap) proposed by EEGLab software. We have chosen permutation as it tests hypotheses concerning distributions, while bootstraps tests hypotheses concerning parameters. In this context, Good et al. 2005 indicated that the bootstrap entails less-stringent assumptions with respect to permutation.

Good, Phillip I. (2005) Permutation, Parametric and Bootstrap Tests of Hypotheses, 3rd ed., Springer ISBN 0-387-98898-X.

4) For clarity purposes, we think that results should be organized by conditions. For every conditon, the occipital (including Pz, P3, P4, Oz, O3, O4), central (Cz, C3, C4) and frontal (Fz, F3, F4) scalp areas involvements are discussed.

5) We have modified the discussion and compared our results with the related litterature (Yodanova et al. 2006, Lenz et al. 2008, 2010, Lenartowicz et al. 2014, Kim et al. 2015). Notably we have discussed the work of Silberstein et al. (2016, 2017).

We have taken into account all these constructive suggestions. We hope that after the modification that we have made, the paper can be now considered for publication.

Round 2

Reviewer 2 Report

The authors have addressed all my comments satisfactorily and the paper can be considered for publication.